# Synthesis of Titanium Oxide Nanotubes Loaded with Hydroxyapatite

**DOI:** 10.3390/nano13202743

**Published:** 2023-10-11

**Authors:** Elinor Zadkani Nahum, Alex Lugovskoy, Svetlana Lugovskoy, Alexander Sobolev

**Affiliations:** Department of Chemical Engineering, Ariel University, Ariel 4070000, Israel; elinorna@ariel.ac.il (E.Z.N.); lugovsa@ariel.ac.il (A.L.); svetlanalu@ariel.ac.il (S.L.)

**Keywords:** anodizing, titanium oxide nanotubes, growth kinetics, corrosion resistance, hydroxyapatite, biomedical applications

## Abstract

A simple method of synthesis of TiO_2_ nanotubes (TiNT) loaded with hydroxyapatite (HAP) is described. Such nanotubes find wide applications in various fields, including biomedicine, solar cells, and drug delivery, due to their bioactivity and potential for osseointegration. The Cp-Ti substrate was anodized at a constant voltage of 40 V, with the subsequent heat treatment at 450 °C. The resulting TiNT had a diameter of 100.3 ± 2.8 nm and a length of 3.5 ± 0.04 μm. The best result of the growth rate of HAP in Hanks’ balanced salt solution (Hanks’ BSS) was obtained in calcium glycerophosphate (CG = 0.1 g/L) when precipitates formed on the bottom and walls of the nanotubes. Structural properties, surface wettability, corrosion resistance, and growth rate of HAP as an indicator of the bioactivity of the coating have been studied. X-ray diffraction (XRD), scanning electron microscope (SEM), energy dispersive spectroscopy (EDS), potentiodynamic polarization test (PPC), electrochemical impedance spectroscopy (EIS), and contact angle (CA) measurements were used to characterize HAP-loaded nanotubes (HAP-TiNT). The CA, also serving as an indirect indicator of bioactivity, was 30.4 ± 1.1° for the TiNT not containing HAP. The contact angle value for HAP-TiNT produced in 0.1 g/L CG was 18.2 ± 1.2°, and for HAP-TiNT exposed to Hanks’ BSS for 7 days, the CA was 7.2 ± 0.5°. The corrosion studies and measurement of HAP growth rates after a 7-day exposure to Hanks’ BSS confirmed the result that TiNT processed in 0.1 g/L of CG exhibited the most significant capacity for HAP formation compared to the other tested samples.

## 1. Introduction

New advances in medicine related to implant prosthetics have become necessary in light of a growing elderly population and the spread of osteoporosis, a condition where bone density deteriorates, increasing the risk of fractures [1]. To solve this problem, polymer, ceramic, and metal materials have been widely used, which can replace damaged or lost bone matrices in the human body. Stainless steel, Co-Cr alloys, and Ti alloys are commonly used as metal materials compatible with the human body due to their impressive tensile strength, fatigue strength, and fracture toughness [2,3,4]. With all their outstanding mechanical properties, stainless steel and Co-Cr alloys have some disadvantages. These include a decrease in fatigue strength and an increase in cytotoxicity, which result from the interaction with biological fluids in the body and the formation of corrosion products from the implanted material. This, in turn, reduces the material’s ability to maintain biocompatibility with cells and tissues of the body [5]. Unlike steel and Co-Cr alloys, Ti alloys demonstrate excellent biocompatibility [6,7,8], high fatigue strength, relatively low density, and good corrosion resistance [9]. However, integrating titanium alloy parts with the living tissue leaves much to be desired, even if threads or rougher surfaces are made on the metal implant.

Electrochemical anodizing is a simple, inexpensive, and readily controllable approach to address this issue. This method allows the production of TiNT with controlled diameters and homogeneous structures, contributing to enhanced biocompatibility of metal implants, improved osseointegration, and a larger surface area for bonding with bone cells [10,11].

The main factors that impact the morphology of nanotubes are temperature, electrolyte composition [12,13,14,15], and anodization voltage [16]. Decreasing the anodization temperature from 35 °C to 0 °C increases the wall thickness and length of the nanotubes [17]. Generally, room temperature (25 °C) is considered optimal for anodizing [18].

The electrical parameter of the anodizing process is another critical parameter influencing the diameter and length of the nanotubes. Linearly increasing the voltage directly affects the diameter and size of the produced nanotubes [19].

The resulting TiNT structures are an excellent basis for loading inside various mineral compounds, significantly improving their physical, chemical, and biological properties [20]. For example, introducing silver into a nanotubular structure [21] enhances the antibacterial properties of the surface. A. Esmaeilnejad et al. incorporated manganese oxide particles into a TiNT structure [22], significantly increasing the surface’s bioactivity. The bone matrix contains many chemical elements, such as Mn, Sr, Ca, P, and others [23]. The insertion of Ca and P into the surface of implantable materials significantly enhances their biocompatibility, bioactivity, and ability to fuse with bone tissue without causing any cytotoxic effects. A study by R. Ramadoss et al. [24] supports the importance of calcium and phosphorus for bone regeneration procedures. Another investigation carried out by A. Schwartz et al. [25,26] involves coating a Ti-6Al-7Nb substrate with an oxide-ceramic layer produced by micro-arc oxidation, consisting of Ti oxides and HAP and having improved biological characteristics of the surface.

However, loading HAP into a TiNT structure is a complex and multi-stage process requiring all stages’ control and compliance. For example, in the work of X. Shen et al. [27,28], the loading process of hydroxyapatite involved a multi-stage process, including soaking and washing the samples. Several methods for loading hydroxyapatite use combined methods [29], such as UV treatment of organic precursors of calcium and phosphorus with their decomposition and formation of HAP. Also quite common are methods of electrochemical treatment [30,31] of the TiNT structure in electrolytes containing calcium and phosphorus precursors to fill the nanotubular structure with them. The methods used are complex in technological execution and require strict adherence to all technological parameters. Using the above methods does not allow controlling the degree and type of filling of the TiNT structure with hydroxyapatite, which can reduce the total contact surface of the implant with biological solutions, thereby reducing its osseointegration properties.

The performed research makes it possible to load HAP into the TiNT structure using a simple technological method of soaking in a one-component electrolyte, with the ability to control the type and degree of filling of the TiNT structure depending on the concentration of CG. The Cp-Ti samples with a TiNT surface layer were immersed in CG solutions (0.1–10 g/L), forming nucleation centers for HAP (promoting bone tissue growth) inside the nanotubular structure. This result is presented for the first time and has no analogs in the literature.

The structure and properties of the resulting coating were comprehensively studied by XRD, SEM, and CA. The growth rate of the HAP coating, serving as an indicator of bioactivity, was assessed through an exposure test in Hanks’ BSS. Electrochemical properties were studied using PPC and EIS.

Based on the study performed, a possible mechanism for the formation of HAP in TiNT is presented.

## 2. Materials and Methods

### 2.1. Experimental Procedures

In this study, TiNTs were synthesized on Cp-Ti (Holland Moran Ltd., Yahud, Israel, 99.7 wt. % Ti, and 0.3 wt. % Fe). The specimens were cleaned and degreased with acetone in an ultrasonic bath.

To reduce the surface roughness of the samples, electrochemical polishing was performed using a mixture of acids (99.5 wt. % acetic acid and 65 wt. % perchloric acid, Rhenium, Modi’in, Israel) in a volume ratio of 9:1. The MP2-AS 35 power supply (Magpulls, Sinzheim, Germany) was utilized as the power source, operating in the following voltage pulse mode: 40 V for 10 s followed by 60 V for the next 10 s. Electrochemical polishing was performed for 5 min at 25 °C with constant stirring (300 rpm). The total polished area of a specimen was 1 cm^2^. The electrochemical cell was made of Teflon with a water-cooled circuit and a fixed distance between the electrodes L = 3 cm. The cathode material is Ti (99% Alfa Aesar, MA, USA).

The anodizing process used the same electrochemical cell, power source, electrode spacing, cathode material, stirring speed, and electrolyte temperature.

A TiNT-containing oxide layer was produced in ethylene glycol containing 0.3 wt. % NH_4_F, 0.66 wt. % CH_3_COONa, and 2 wt. % H_2_O [32]. The chemical reagents used were purchased from Alfa Aesar, Massachusetts, USA. The electrolyte solution was prepared by adding the calculated amount of water to NH_4_F, and then the resulting mixture was added to CH_3_COONa, previously dissolved in ethylene glycol. The resulting mixture was stirred (300 rpm) on a magnetic stirrer for an hour to obtain a homogeneous electrolyte composition. The anodizing process lasted 2 h at 40 V. The treated samples were washed in distilled water and dried.

To achieve a crystalline structure and enhance the bioactive properties of the surface, the samples were treated in a Carbolite CWF 13/13 furnace (Carbolite Gero Ltd., Hope Valley, UK) in an air atmosphere. The heating rate was set at 5 °C/min until reaching a temperature of 450 °C, followed by an isothermal exposure at 450 °C for 2 h [33]. The samples were cooled together in a furnace. Thus, treated samples were soaked in a solution of CG (C = 0.1; 10 g/L) with a magnetic stirrer at a constant stirring speed (300 rpm) for 24 h.

The processes of preparing TiNT and loading it with HAP are shown in Figure 1.

To study the bioactivity of the resulting coatings, the samples were exposed to Hanks’ BSS (Sigma-Aldrich, St. Louis, MO, USA) for 0 and 7 days. Each sample was immersed in 20 mL of the solution and placed in a closed thermostatic bath at 37 ± 0.5 °C. The solution was discarded every other day, and a fresh portion of this solution was added instead to keep the ion concentration constant.

### 2.2. Analytical Methods

Phase identification of prepared samples was performed using a Cu-Ka gun (λ = 1.54059 Å) using an XRD-SmartLab SE instrument (XRD, Rigaku, Tokyo, Japan) at 30 mA, 40 kV, and a scale of 2θ (0.03°/step and scanning speed of 0.3°/min). The International Center for Diffraction Data (ICDD) standards were used as a reference throughout all stages.

The surface morphology and elemental composition of samples were characterized by a scanning electron microscope (SEM, TESCAN MAIA3 Triglav™, Brno–Kohoutovice, Czech Republic) equipped with an energy dispersive spectroscopy analyzer (EDS, Aztec Oxford Instruments, Concord, MA, USA).

Surface wettability characteristics were assessed through contact angle measurements using the sessile drop technique with a volume of 6 μL. Hanks’ BSS (Sigma-Aldrich, St. Louis, MO, USA), representing a polar liquid, and glycerol (Sigma-Aldrich, St. Louis, MO, USA), a non-polar liquid, as test fluids. The Owens–Wendt method [34] was employed to calculate the surface energy (γ) based on the parameters of these distinct liquids. The experiments were conducted under ambient conditions at a temperature of 25 °C, and the results were based on the mean value obtained from 5 measurements.

For the electrochemical experiments, PPC and EIS were conducted using a PARSTAT 4000A instrument (Princeton Applied Research, Oak Ridge, TN, USA) in Hanks’ BSS. The chemical composition of Hanks’ BSS is shown in Table 1. The corrosion resistance of samples S0–S5 was studied at a human body temperature of 37 ± 0.5 °C.

An electrochemical three-electrode cell arrangement was implemented for the experiments. The reference electrode was Ag/AgCl with a 3.5 M KCl electrolyte; the counter electrode was a platinum sheet. To achieve OCP stabilization, the samples underwent one hour of immersion in Hanks’ BSS before testing.

The experiment involved recording PPCs at a potential range of ±250 mV relative to the OCP, and the scanning rate used was 0.1 mV/s. Additionally, EIS data were collected at the OCP, spanning a wide frequency range from 100 kHz to 0.1 Hz, with a Root Mean Square (RMS) amplitude of 5 mV.

The data analysis was performed using VersaStudio, developed by AMETEK Scientific Instruments in Kingston, UK, in conjunction with EC-Lab^®^ software V11.10, provided by Biologic Science Instruments in Seyssinet-Pariset, France.

The experimental samples have been assigned codes, presented in Table 2, along with their corresponding treatments.

### 2.3. Statistical Analysis

Because of the purely descriptive philosophy of this study, no statistic model could be formulated in terms of “*p*-value”, “null hypothesis”, etc. The only statistics used were the determinations of the dispersion of the measurement results based on the mean squared deviations from the average for each series of repeated measurements.

## 3. Results and Discussions

### 3.1. Anodizing Process

Experimental anodizing conditions are essential for forming well-aligned TiNT arrays with a given pore size and wall thickness. The formation of TiNT arrays in a fluoride electrolyte is the result of two simultaneous processes: (a) oxidation of Ti at the metal/oxide interface (reactions 1 and 2) and (b) chemical dissolution of the barrier layer at the TiO_2_/electrolyte interface, leading to the formation of a tubular structure (reactions 3 and 4). The key point in the formation and growth of TiNT is the rate of competing reactions between mechanisms (a) and (b). In general, the anodizing process can be represented by the following sequence of reactions [35]:Ti → Ti^4+^ + 4e^−^(1)
Ti^4+^ + 2H_2_O → TiO_2_ + 4H^+^(2)
Ti^4+^ + 6F^−^ → [TiF_6_]^2−^(3)
TiO_2_ + 6F^−^ + 4H^+^ → [TiF_6_]^2−^ + 2H_2_O(4)

Figure 2 shows current transients recorded during anodizing at a voltage of 40 V. A characteristic exponential decrease in current is observed during the initial minutes of anodization. This phenomenon is explained by the competing processes of forming a barrier layer and the nucleation of the TiNT structure. Since the current decreased exponentially in the first minutes of the process, we can assume that the predominant process in the selected period was forming a barrier layer (reactions 1 and 2) of Ti oxide. The end of the formation of the barrier layer and the beginning of the formation of TiNT (reactions 3 and 4) can be considered the time interval when the current reaches a quasi-stationary value. At the applied potential, the quasi-stationary value of the current corresponds to 1.4 mA.

The resulting current curve is typical for electrolytes based on ethylene glycol. The high viscosity of ethylene glycol significantly affects the diffusion of substances into the reaction zone, slowing down the chemical and electrochemical dissolution of the titanium oxide barrier layer. Reducing the dissolution rate of the barrier layer is the key to forming a more ordered TiNT structure.

### 3.2. Morphology

Figure 3 illustrates typical SEM images captured from CP-Ti samples after anodization at a potential of 40 V.

The samples obtained after anodizing and high-temperature annealing had a TiNT coating structure with an outer pore diameter of 100.3 ± 2.8 nm and a wall thickness of about 17.9 ± 0.2 nm. The length of the formed tubular structure is 3.5 ± 0.04 µm. The chemical analysis of the EDS surface (Table 3, Spectrum 1 (Sp. 1)) and cross-section (Table 3, Sp. 2) of the obtained sample shows an almost stoichiometric presence of Ti and O.

Figure 4 shows the morphology of samples S2 and S3 after their exposure to CG solutions with a concentration of 0.1 g/L (a,b) and 10 g/L (c,d), respectively. The samples were exposed for 24 h.

Based on the results of EDS point chemical analysis for samples S4 and S5, it was determined that the outer part of the pore walls (Sp. 3 and Sp. 5) consists of Ti and O in nearly stoichiometric amounts. The inner part of the pores of samples S4 (Figure 4b, Table 3, Sp. 4) and S5 (Figure 4d, Table 3, Sp. 6) consists of a mixture of Ti, O, Ca, and P. The ratio of Ca and P forming a porous coating inside the nanotubular structure of titanium oxides has a Ca/P ratio in the range of 1.60–1.74 (Table 3), which may correspond to the chemical composition of calcium hydroxyapatite (Ca/P = 1.67) [36].

The formed HAP on the walls of tubular pores (Figure 4a,b) and inside them (Figure 4c,d) is a hydrolysis product of calcium glycerophosphate according to the following reactions:C_3_H_7_CaO_6_P + H_2_O → CaHPO_4_ + C_3_H_5_(OH)_3_(5)
7CaHPO_4_ + H_2_O → Ca_5_(PO_4_)_3_OH + 2Ca(H_2_PO_4_)_2_(6)

The hydrolysis of CG proceeds in two steps (reactions 5 and 6). The first step (reaction 5) is the formation of calcium hydro-orthophosphate and glycerol. In the second step (reaction 6), calcium hydro-orthophosphate is hydrolyzed to HAP and calcium dihydro-phosphate. HAP is an insoluble compound that precipitates inside the TiNT due to stepwise hydrolysis (Figure 4). The second hydrolysis product, calcium dihydrogen phosphate, is a soluble compound dissociated into ions.

The thickness of the formed HAP on the pore walls of sample S2 is about 10 nm and has a nanoporous structure.

With an increase in the concentration of CG to 10 g/L (S3), during soaking, the pores were filled with HAP, which is its less preferred form, since the primary purpose of HAP introduced into the pores is, on the one hand, to increase the surface contact area with biologically active liquids and, on the other hand, to increase the biological activity of the TiNT structure.

To indirectly determine the bioactivity of the obtained surfaces, these coatings were exposed to Hanks’ BSS for 7 days at 37 ± 0.5 °C (S4 and S5). The results of the morphology and phase composition evolution are presented in Figure 5 and Figure 6, respectively.

Figure 5a shows that after 7 days of exposure to Hanks’ BSS, the formation of an amorphous-crystalline structure of HAP occurred on sample S4, which is also confirmed by EDS elemental analysis (Sp. 7, Table 3) and XRD phase analysis (Figure 6d). It should also be noted that hydroxyapatite formation on the sample surface in such a short time may indicate its high bioactivity and osseointegration properties. For sample S5, the amorphous-crystalline structure of HAP was not formed, or the rate of its formation was very low. This trend is confirmed by the surface morphology (Figure 5b) and the chemical analysis of EDS (Sp. 7, Table 3) after 7 days of exposure to Hanks’ BSS. Therefore, filling pores with HAP as centers of crystallization and growth of bone neoplasms is a less preferable procedure than its formation on the walls of nanotubes.

### 3.3. XRD Analysis

Figure 6 shows the results of phase analysis of S0 after anodizing and heat treatment (S1), precipitation of HAP (S2) in a solution of CG (C = 0.1 g/L), and exposure to Hanks’ BSS for 7 days (S4).

To assess the phase composition, a series of measurements were carried out on an untreated Cp-Ti (S0) after anodizing and annealing on air for 2 h at a temperature of 450 °C (S1), after soaking in a solution of CG = 0.1 g/L (S2), and after exposure to Hanks’ BSS for 7 days (S4). The Cp-Ti sample (S0) consists of the characteristic phases of titanium metal (ICDD 00-064-0863). After the process of anodizing and annealing, characteristic peaks of anatase with angles of 2θ—25°, 37.8°, 48°, 54°, 55°, 69°, and 75.2° were visible, indicating that the amorphous tubular structure has been transformed into the crystalline phase of anatase (Figure 7). The first anatase peak (ICDD 01-083-4054) corresponding to the (101) plane is intense and appears at 2 θ ≈ 25°. The presence of a peak at 25° (anatase ICDD 01-083-4054) and the absence of a peak at 27.45° (rutile) indicate that the nanotubes consist entirely of the anatase phase (Figure 7).

After soaking the sample in a solution of CG and its slow hydrolysis, hydroxyapatite (ICDD 00-064-0738) was formed on the surface of the TiNT structure (Figure 4a,b; Figure 8), which is confirmed by its characteristic peaks in the X-ray diffraction pattern Figure 6 c at angles 2θ—25.8°, 31.76°, 32.14°, 32.90°, as well as by EDS analysis carried out earlier in Section 3.2. During the subsequent exposure of the bioactive coating in Hanks’ BSS (T = 37 ± 0.5 °C, 7 days), the phase composition does not undergo significant changes (Figure 6d), which is associated with the formation of an amorphous-crystalline HAP layer (Figure 8), which is also confirmed by the surface morphology (Figure 5a). A similar phase composition was obtained on samples S3 and S5 before and after exposure to Hanks’ BSS, respectively.

Based on the XRD analysis results, a semi-quantitative analysis was performed, and its results are presented in Figure 7.

### 3.4. Contact Angle

The attainment of sufficient initial implant stabilization is a critical element for successfully integrating implants with bone tissue [37]. This primary stability is influenced by the implant’s morphology and shape factors [38]. The presence of a porous structure and a bioactive material increases its contact area with the biological fluid and accelerates the formation of the bone matrix. The latter’s formation is associated with the efficient attachment and proliferation of structural proteins and glycoproteins to the implant surface [39]. The rate at which the proliferation process occurs is directly influenced by the hydrophilic characteristics of the surface and its energy. To investigate the hydrophilic properties and surface energy of the samples, S0–S5 were examined, and their CAs were measured using Hanks’ BSS at 37 ± 0.5 °C and glycerol at 25 ± 0.5 °C. The results of the CA and surface energy are illustrated in Figure 8 and summarized in Table 4.

Sample S0 had a less hydrophilic surface with CA = 78.2 ± 1.5° compared to other samples in Table 4.

After electrochemical polishing, anodizing, and heat treatment of the sample surface, its contact angle decreased to 30.4 ± 1.1°, almost 2.5 times less than that of the untreated metal. On the sample S2 treated during the day in a solution of CG, a nanoporous structure formed on the walls of TiNT with a decrease in the contact angle to 18.2 ± 1.2°. With an increase in the concentration of CG (S3), the contact angle increased to 24.5 ± 1.2°, most likely due to the filling of the TiNT structure and the effect on its hydrophilic properties of the coating. During exposure to Hanks’ BSS, amorphous HAP was formed (Figure 5), which, in the case of sample S4 (Figure 5a), completely covered the surface with the formation of a uniform layer, thereby reducing the contact angle to 7.2 ± 0.5°. The resulting coating on sample S5 has higher contact angles and appears (Figure 5b) to have lower bioactivity. Similar patterns of surface response are observed for SA in glycerol, but a slight increase in hydrophobicity was observed for the latter.

The results in Figure 8 and Table 4 regarding surface energy (γ) are consistent with the results for CA. In other words, the surface energy varies inversely with the contact angle. Therefore, the higher the surface energy, the smaller the contact angle. After analyzing Figure 7 and Table 4, it can be noted that the lowest energy value of 45.2 ± 2.2 [mJ ∙ m^−2^] was measured for the substrate that had the highest contact angle. This energy value increases and reaches a maximum of 485.3 ± 1.5 [mJ ∙ m^−2^] for sample S4, corresponding to the smallest contact angle.

The decrease in contact angles and subsequent increase in surface energy, both in nanotubes and nanopores, are caused by liquid penetration facilitated by capillary forces [40]. Surface energy and contact angle directly relate to protein adsorption in cell culture media and body fluids. Therefore, the higher the surface energy (γ) and the lower the contact angle, the greater the adhesion and the higher the rate of osseointegration [41].

### 3.5. Electrochemical Test

The corrosion behavior of the samples (Table 5) was analyzed using the PPC test and EIS in Hanks’ BSS at 37 ± 0.5 °C. PPC plots are presented in Figure 9, and the corresponding design parameters are given in Table 5.

The corrosion potentials (E_corr_), corrosion current densities (I_corr_), and slopes of the linear sections corresponding to the anodic and cathodic processes (β_a_ and β_c_) were assessed from the appropriate polarization curves in Tafel coordinates. The polarization resistance values (R_p_) were then calculated using the Stern-Geary equation [42].
(7)$$Rp=\frac{\beta_{a}\times \beta_{c}}{2.3\times i_{corr}\times (\beta_{a}+\beta_{c})}$$

According to the obtained parameters (E_corr_, I_corr_, β_a_, β_c_) and calculated polarization resistance (Rp), sample S0 exhibits the highest susceptibility to corrosion. Additionally, the polarization resistance value for sample S0 is determined to be 13.77 [kΩ ∙ cm^−2^].

Samples S1–S3 show a decreasing trend in corrosion currents, leading to an increase in polarization resistance in the following sequence: S1 < S3 < S2. The corresponding polarization resistance values for these samples are 72.79 [kΩ ∙ cm^−2^], 79.82 [kΩ ∙ cm^−2^], and 83.91 [kΩ ∙ cm^−2^], respectively. This trend is directly related to sample processing methods, namely, anodizing which promotes the formation of a TiNT coating with an ordered structure, and heat treatment (Figure 6b) which enables the phase transition of TiNT from an amorphous structure to crystalline anatase. A further increase in polarization resistance and a decrease in corrosion currents are associated with the encapsulation of coating defects with HAP formation on the walls of the tubular matrix of TiNT (S2) or its filling in the case of sample S3.

The EIS method was used to assess the corrosion properties and the growth of the HAP structure in samples S2–S5 during 7 days of exposure to Hanks’ BSS. The equivalent electrical circuit (EEC) used to model the process of corrosion and HAP growth is shown in Figure 10. The interfaces between the electrolyte and the electrode comprise an electrolyte/TiNT layer and an electrolyte/barrier layer. The circuit elements R_s_, R_1_, CPE_1_, R_2_, and CPE_2_ represent the various components in this EEC.

Specifically, Rs denotes the electrolyte resistance, which is measured to be 37 ± 2 Ohms. Rs is connected in series, with R_1_ corresponding to the electrolyte resistance within the pores and CPE_1_ representing the capacitance of the TiNT layer with non-ideal geometry. The R_2_–CPE_2_ couple describes the charge transfer process at the barrier layer/electrolyte interface and represents the resistance to charge transfer (R_2_) and the capacitance of the electrical double layer (CPE_2_) [43].

The selection of this EEC in Figure 10, along with the electrical elements (R_s_, R_1_, CPE_1_, R_2_, CPE_2_), fits well with the observed morphology of the cross sections of the samples shown in Figure 4, which exhibit TiNT and barrier layer structure.

Impedance spectroscopy spectra [44].

Considering the non-ideal behavior of the simulated interface, the capacitance can be represented by a Constant Phase Element (CPE), which is described by the following equation [45]:(8)$$Z_{CPE}=\frac{1}{Y_{0}{\left(j\omega \right)}^{n}}$$
where Z_CPE_ is the impedance of the CPE, Y_0_ is the admittance of an ideal capacitance, j is the imaginary unit (√−1), ω is the angular frequency (2πf), and n is a dimensionless parameter that varies from 0 to 1. At n = 0, CPE behaves like a resistor, and at n = 1, CPE behaves like an ideal capacitor.

The Nyquist and Bode plots for samples S2–S5 are shown in Figure 11; the design parameters are shown in Table 6.

The experimental curves are in good agreement with the fitting results (Table 6), as evidenced by the chi-square (χ^2^) values not exceeding 10^−3^. This confirms the appropriateness of the equivalent electrical circuit (EEC) choice and the accuracy of the approximation level between the experimental curves and the results of spectrum simulation using the EC-Lab V11.10 program.

The fitting of the Nyquist plot for samples S2–S5 resulted in depressed semicircles, as shown in Figure 11a,b. The appearance of a depressed semicircle on a Nyquist plot often suggests the presence of two capacitive loops with comparable time constants. The presence of these two capacitive loops is further emphasized by the significant increase in phase angles observed in the Bode diagrams, as illustrated in Figure 11c. This phase angle growth is a typical trait of a two-layer configuration, where the outer layer corresponds to a surface nanotubular structure and the inner layer acts as a dense barrier.

In the Nyquist plot, the high-frequency capacitance loop corresponds to the combined effect of the capacitance of the TiNT layer (CPE_1_) and the electrolyte resistance in the pores (R_1_). Conversely, at medium and low frequencies, a low-frequency loop emerges (CPE_2_), which can be attributed to the capacitance of the electrical double layer at the interface between the barrier layer and the metal substrate, along with the charge transfer resistance (R_2_) [46].

A more detailed analysis of the frequency dependence on the phase angle (Figure 11c) and total impedance (Figure 11d) for samples S2 and S3 at the initial time of exposure shows the presence of two kinks in the high-frequency and mid-frequency regions, which correspond to complex two-layer structure (Figure 4) samples. The resistance (R_1_) and capacitance characteristics (CPE_1_) of the TiNT layer for both samples have similar values of 0.71–0.74 (kΩ·cm^2^) and (7.05–6.54) 10^−6^ (S·cm^−2^·s^− n^), respectively. However, the barrier layer has minor differences, such as the charge transfer resistance (R_2_) of sample S2 = 83.64 (kΩ·cm^2^), which is slightly higher than that of sample S3 = 78.90 (kΩ·cm^2^). An increase in the parameter R_2_ and a slight decrease in the capacitance properties of CPE_2_ for sample S2 is associated with the encapsulation of barrier layer defects during the formation of a porous HAP structure at the bottom and walls of the tube (Figure 4a,b). The capacitive properties of samples S2 and S3 have similar values for the nanotubular layer n_1_ = 0.73–0.76 and for the barrier layer n_2_ = 0.97–0.98. However, it should be noted that the barrier layer has a more pronounced capacitive nature than the TiNT layer. It should also be noted that samples S2 and S3 at zero exposure time in Hanks’ BSS have similar characteristics of the impedance modulus Z_f→0 Hz_ = 84.39 (kΩ·cm^2^) and Z_f→0 Hz_ = 79.61 (kΩ·cm^2^), respectively. This result is in good agreement with the results of the polarization resistance (R_p_) obtained by the Tafel linear extrapolation method and amounts to S2 = 83.91 (kΩ·cm^−2^) and S3 = 79.82 (kΩ·cm^−2^) for the sample.

During the 7-day exposures of samples S4 and S5 to Hanks’ BSS, the high-frequency peaks shift to a lower-frequency region, indicating a change in the morphology of the surface layer, specifically its compaction. This compaction may be attributed to the formation of an amorphous hydroxyapatite (HAP) layer on the surface of the samples, a phenomenon further confirmed by an increase in their impedance modulus and a change in surface morphology (see Figure 5). A more detailed analysis of samples S2–S5 shows (Table 6) that there is a decrease in the capacitive characteristics of the TiNT coating (CPE_1_) after 7 days of exposure (7.05–1.95) 10^−6^ (S·cm^−2^·s^−n^), indicating a slowdown in the rate of corrosion processes. The value of the dimensionless coefficient n1 for samples S2–S5 is in the range of 0.73–0.85, which indicates the capacitive nature of the simulated CPE_1_ element.

In the case of the barrier layer, a similar trend is observed, i.e., during exposure to Hanks’ BSS, the charge transfer resistance increases. For example, for samples S2 and S4, it grows from 83.64 (kΩ∙cm^2^) to 104.95 (kΩ·cm^2^), while for S3 and S5 from 78.90 to 82.94 (kΩ·cm^2^). With an increase in the charge transfer resistance, the capacitance of the electrical double layer naturally decreases for samples S2 and S4 (2.25–4.13) 10^−6^ (S·cm^−2^·s^−n^), while S3 and S5 (3.22–3.54) 10^−6^ (S·cm^−2^·s^−n^). The CPE_2_ element describes the behavior of the barrier layer under these conditions, and the value of the dimensionless coefficient n_2_ tends to be 1 (Table 6), which is, by its nature, an almost ideal capacitor.

In addition, it should be noted that sample S2 (Figure 11c) at zero exposure time had two kinks (low and medium frequencies), and after 7 days of exposure, sample S4 had one kink at medium frequencies. That may indicate the formation of a more uniform surface layer, which is also confirmed by morphology studies (Figure 5a) and the highest modulus of impedance Z_f→0 Hz_ = 106.7 (kΩ·cm^2^).

Based on the information provided earlier, we can infer that the presence of a TiNT layer loaded with HAP is responsible for expediting the osseointegration process, particularly aiding in the nucleation and growth of HAP. The dense barrier layer is responsible for the anti-corrosion properties since R_2_≫R_1_ (Table 6).

## 4. Conclusions

In this research, a well-arranged TiNT structure of TiO_2_ was successfully produced on the surface of a titanium implant through a two-step process involving electrochemical anodizing and subsequent heat treatment. The resulting nanotubes have an average diameter of 100.3 ± 2.8 nm and a length of 3.5 ± 0.04 µm. To enhance bioactivity and structural properties, the titanium nanotube (TiNT) structure was loaded with hydroxyapatite (HAP), obtained by immersing samples for 24 h in solutions of calcium glycerophosphate (CG) at concentrations of 0.1 g/L and 10 g/L, followed by its stepwise hydrolysis. SEM analysis of the formed coatings revealed significant morphological differences after exposure to 0.1 g/L and 10 g/L CG. In the case of the sample treated in a solution (0.1 g/L), a nanoporous structure of HAP was formed on the walls and at the bottom of TiNT. This type of loading of the TiNT structure had the smallest contact angle before exposure (CA = 18.2 ± 1.2°) in Hanks’ BSS and after it (CA = 7.2 ± 0.5°). The results of the 7-day exposure of the sample to Hanks’ BSS showed that the chosen type of modification and decoration of the sample increased the rates of HAP formation and corrosion properties (Z_f→0 Hz_ = 106.7 kΩ∙cm^2^) due to partial encapsulation of the barrier and TiNT layers, respectively. Based on the obtained results, it was suggested that decorating a TiNT structure with HAP occurs during the stepwise hydrolysis of CG via the following reactions:C_3_H_7_CaO_6_P + H_2_O → CaHPO_4_ + C_3_H_5_(OH)_3_(9)
7CaHPO_4_ + H_2_O → Ca_5_(PO_4_)_3_OH + 2Ca(H_2_PO_4_)_2_(10)

From the information provided earlier, we can infer that the presence of a TiNT layer loaded with HAP should support the osseointegration process, specifically aiding in the nucleation and growth of HAP, which is one of the indicators of surface bioactivity.

## Figures and Tables

**Figure 1 nanomaterials-13-02743-f001:**
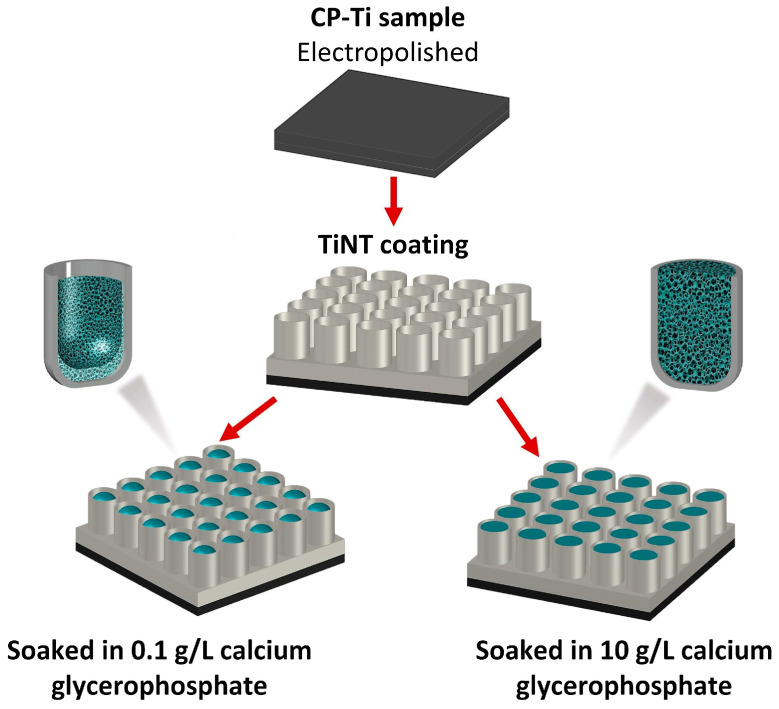
Stages of obtaining titanium oxide nanotubes loaded with hydroxyapatite.

**Figure 2 nanomaterials-13-02743-f002:**
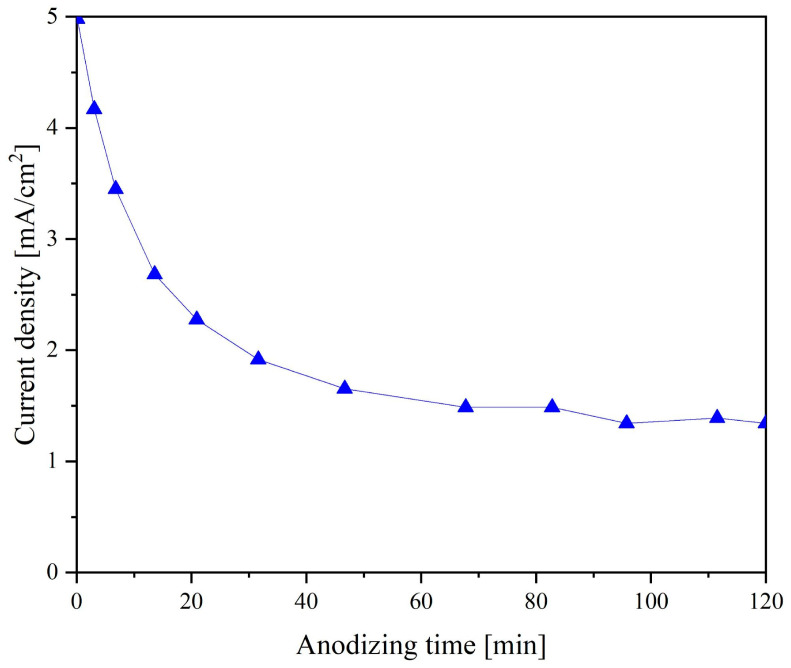
Current densities during anodic growth of barriers and nanotubes at constant voltage (40 V) anodization.

**Figure 3 nanomaterials-13-02743-f003:**
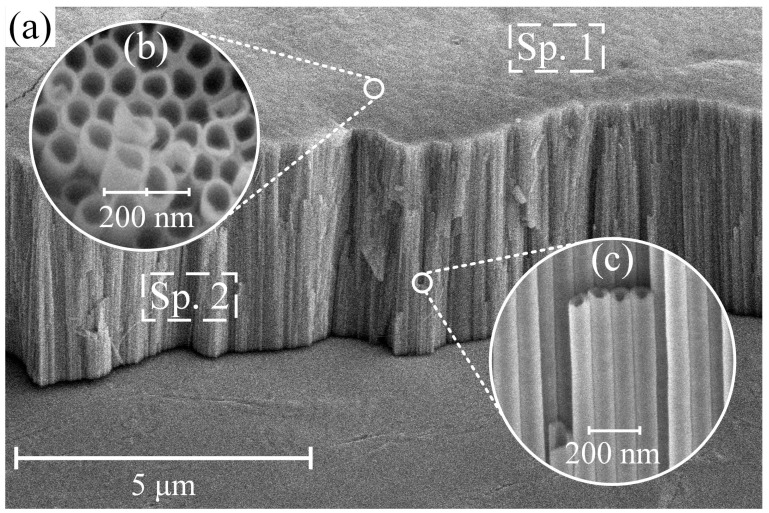
Typical morphology (**a**,**b**) and cross-section (**c**) of sample S1 after anodizing at 40 V and high-temperature annealing for 2 h.

**Figure 4 nanomaterials-13-02743-f004:**
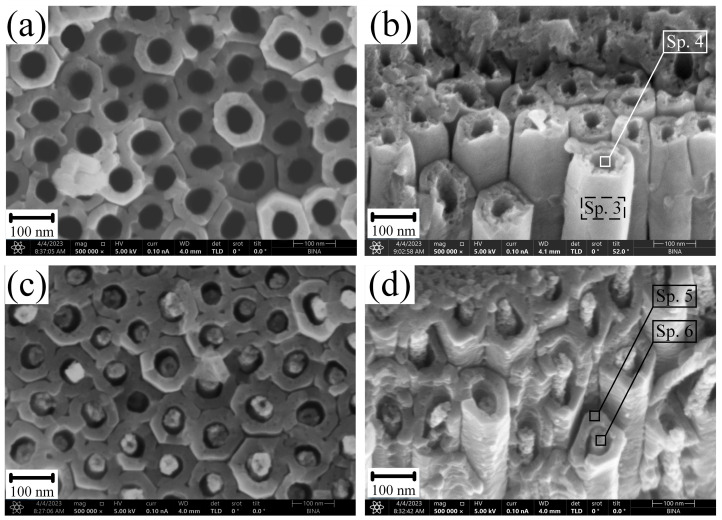
Surface morphology of samples S2 and S3 after 24 h of exposure to calcium glycerophosphate solutions with concentrations of 0.1 g/L (**a**,**b**) and 10 g/L (**c**,**d**).

**Figure 5 nanomaterials-13-02743-f005:**
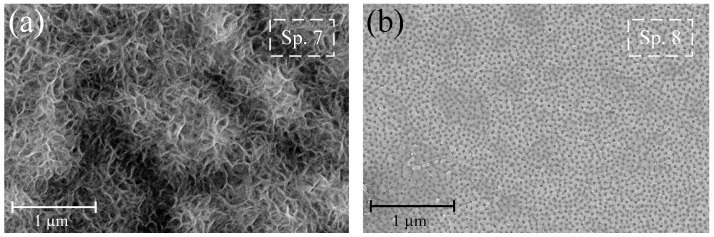
Surface morphology of samples pre-treated in 0.1 g/L (**a**) and 10 g/L (**b**) calcium glycerophosphate solution after 7 days exposure at 37 ± 0.5 °C to Hanks’ balanced salt solution.

**Figure 6 nanomaterials-13-02743-f006:**
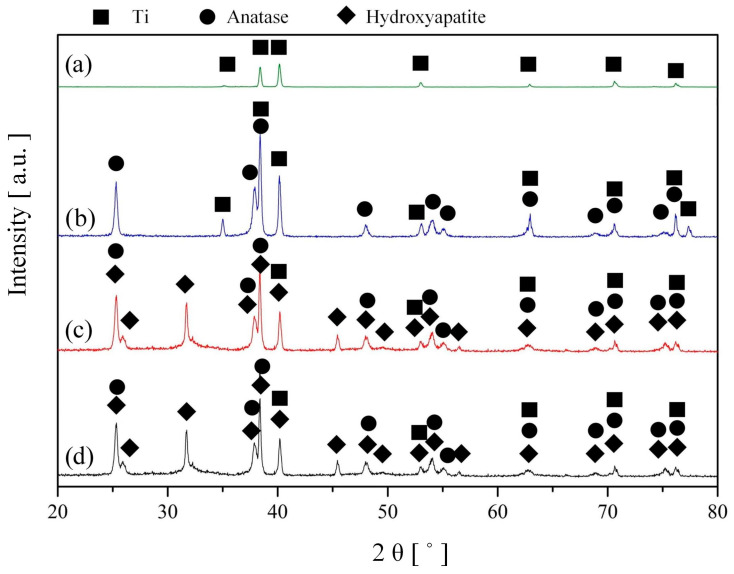
Typical X-ray patterns of (**a**) untreated Cp-Ti, (**b**) after anodizing and heat treatment, (**c**) after soaking of hydroxyapatite in a solution of calcium glycerophosphate (C = 0.1 g/L), and (**d**) exposure to Hanks’ balanced salt solution for 7 days.

**Figure 7 nanomaterials-13-02743-f007:**
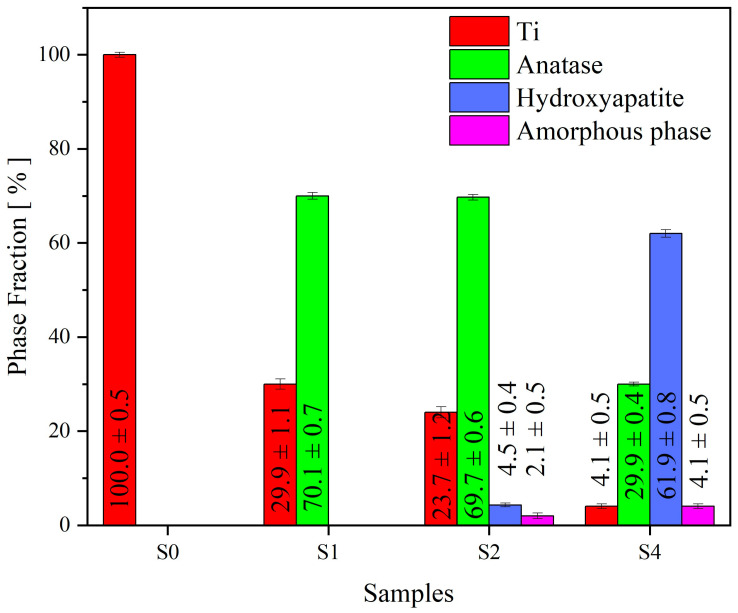
Semi-quantitative analysis of phase fraction of S0–S2, S4 samples.

**Figure 8 nanomaterials-13-02743-f008:**
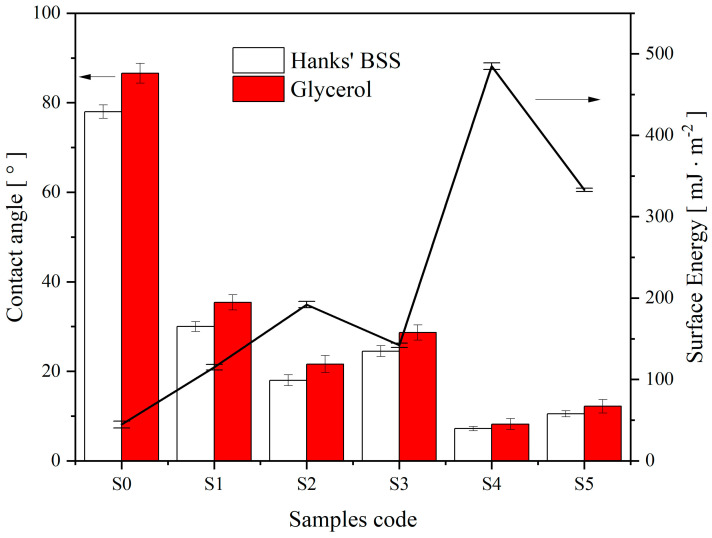
Contact angle and surface energy of the S0–S5 samples were examined with drops of Hanks’ balanced salt solution at 37 ± 0.5 °C and glycerol at 25 °C.

**Figure 9 nanomaterials-13-02743-f009:**
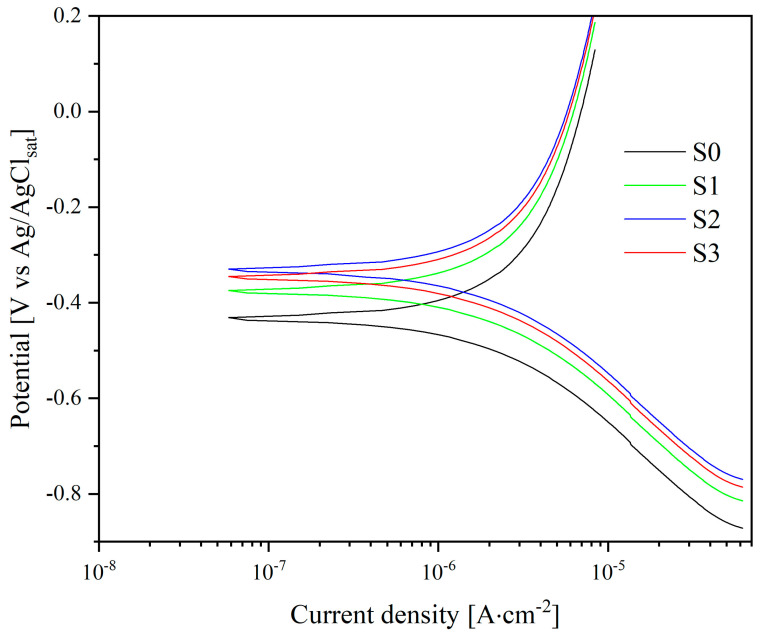
Potentiodynamic polarization test in Hanks’ balanced salt solution at 37 ± 0.5 °C for samples S0–S3.

**Figure 10 nanomaterials-13-02743-f010:**
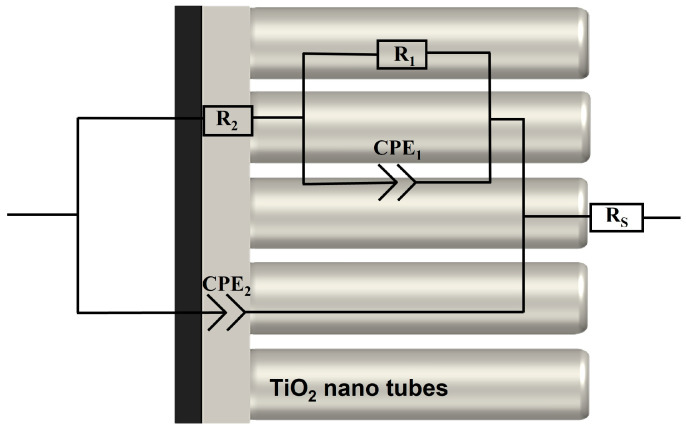
Equivalent electrical circuit for approximation of experimental Electrochemical.

**Figure 11 nanomaterials-13-02743-f011:**
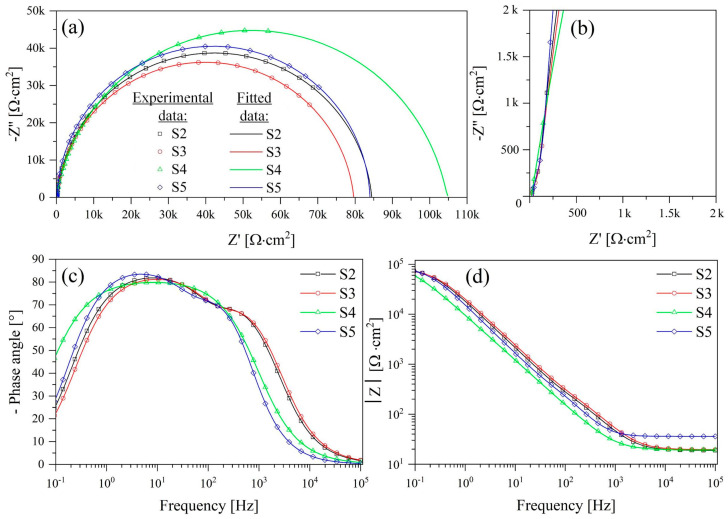
Nyquist and Bode plots for samples S2-S5 after 0 and 7 days of exposure to Hanks’ BSS: (**a**) Nyquist plot; (**b**) low-frequency region of the Nyquist plot; (**c**) phase angle Bode plots; (**d**) Bode impedance modulus plots. Symbols refer to experimental values, and solid lines refer to fitted data.

**Table 1 nanomaterials-13-02743-t001:** Composition of Hanks’ balanced salt solution.

Components	Concentration [g/L]
CaCl_2_	0.185
MgSO_4_	0.09767
KCl	0.4
KH_2_PO_4_	0.06
NaHCO_3_	0.35
NaCl	8.0
Na_2_HPO_4_	0.04788
Glucose	1.0

**Table 2 nanomaterials-13-02743-t002:** Codes of the experimental samples given the relevant treatment.

Samples Code	Anodizing	Heat Treatment	Soaking in Calcium Glycerophosphate Solution		Exposure Test in Hanks’ Balanced Salt Solution	
			C = 0.1 [g/L] 24 h	C = 10 [g/L] 24 h	0 Days	7 Days
S0	Without treatment					
S1	+	+				
S2	+	+	+		+	
S3	+	+		+	+	
S4	+	+	+			+
S5	+	+		+		+

**Table 3 nanomaterials-13-02743-t003:** Energy dispersive spectroscopy elemental analysis data for S1–S5 samples.

Samples Code	Ti [at.%]	O [at.%]	P [at.%]	Ca [at.%]	Ca/P
Sp. 1	30.2	69.8	-	-	-
Sp. 2	32.2	67.8	-	-	-
Sp. 3	32.18	67.56	0.1	0.16	1.60
Sp. 4	29.10	64.57	1.70	2.84	1.67
Sp. 5	32.12	67.50	0.14	0.24	1.71
Sp. 6	27.49	63.00	3.42	5.96	1.74
Sp. 7	-	66.9	13.3	19.81	1.49
Sp. 8	16.8	31.2	6.0	8.4	1.4

Sp.—spectrum of energy dispersive spectroscopy.

**Table 4 nanomaterials-13-02743-t004:** Contact angle and surface energy (γ) values for S0–S5 samples.

Samples Code	Contact Angle [°]		Surface Energy—γ
	Hanks’ BSS	Glycerol	[mJ∙m^−2^]
S0	78.2 ± 1.5	86.6 ± 2.2	45.2 ± 2.2
S1	30.4 ± 1.1	35.4 ± 1.7	115.3 ± 1.4
S2	18.2 ± 1.2	21.6 ± 1.9	192.1 ± 1.6
S3	24.5 ± 1.2	28.7 ± 1.7	142.5 ± 1.5
S4	7.2 ± 0.5	8.3 ± 1.2	485.3 ± 1.5
S5	10.5 ± 0.7	12.5 ± 1.5	332.6 ± 0.9

**Table 5 nanomaterials-13-02743-t005:** Calculated corrosion parameters for samples S0–S3 in Hanks’ balanced salt solution at 37 ± 0.5 °C.

Samples Code	E_corr_ vs. Ag/AgCl_sat_ [mV]	I_corr_ [µA∙cm^−2^]	β_a_ [mV·dec^−1^]	−β_c_ [mV·dec^−1^]	R_p_ [kΩ·cm^−2^]
S0	−428 ± 5	5.02 ± 0.05	218	92	13.77
S1	−373 ± 3	0.872 ± 0.04	228	89	72.79
S2	−343 ± 3	0.832 ± 0.02	221	93	83.91
S3	−352 ± 4	0.840 ± 0.03	222	91	79.82

**Table 6 nanomaterials-13-02743-t006:** Values of the fitted electrochemical empedance spectroscopy parameters obtained from the equivalent electrical circuit for samples S2–S5.

Impedance Spectroscopy Parameters	S2	S3	S4	S5
CPE_1_ × 10^−6^ [S·cm^−2^·s^−n^]	7.05	6.54	1.95	6.20
n_1_	0.76	0.73	0.85	0.79
R_1_ [kΩ·cm^2^]	0.74	0.71	1.76	1.01
CPE_2_ × 10^−6^ [S·cm^−2^·s^−n^]	4.13	3.54	2.25	3.22
n_2_	0.98	0.97	0.95	0.95
R_2_ [kΩ·cm^2^]	83.64	78.90	104.95	82.94
Z_f→0Hz_ [kΩ·cm^2^]	84.39	79.61	106.70	83.95
χ^2^ × 10^−4^	2.7	2.5	3.0	3.6

## Data Availability

All the data supporting the findings of this study are available within the article.

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
