# Peer review of "Synthesis of Titanium Oxide Nanotubes Loaded with Hydroxyapatite"

_nanomaterials, 2023, doi:10.3390/nano13202743_

Round 1

Reviewer 1 Report

The manuscript “Synthesis of titanium oxide nanotubes loaded by hydroxyapatite for biomedical applications” by EZ Nahum et al., can be accepted for publication after the authors properly address all the raised queries (in the order they appear in the manuscript):

1. First of all, the current title of the manuscript is not completely in accordance with the presented results. The authors introduce only the results obtained after the immersion of their fabricated samples in Hanks’ solution. In this Reviewer’s opinion, this is only one of the first steps in demonstrating the capacity of a material to be considered suitable for biomedical applications, as far more complex studies have to be performed. In this respect, the authors should “tame” the title which could read as “Synthesis of titanium oxide nanotubes loaded with hydroxyapatite”. One should note that the authors have the chance to advance such type of application (i.e., biomedical) in the “Conclusions” section or use it as a Keyword, in the dedicated section.

2. The acronyms should be defined the first time they are used in the text. Since some of them do not appear twice in the Abstract (i.e., XRD, SEM, EDS, EIS), they should not be used (page 1, line 19). 

3. In the Introduction section there are many outdated references (which were published 16 – 19 years ago), and the authors must replace them with newer ones.

4. The last paragraph in the “Introduction” section should clearly strengthen both the novelty and the aim of the current research taking into consideration the state of the art in the field. Therefore, the authors have to reconsider the incriminated paragraph.

5. How many “identical” samples were in fact fabricated by the authors for each investigation method? Did they find any statistical significance among the tested groups? In order for these samples to be used for future biomedical applications, it is imperative to have good statistics. In this respect, a special subsection entitled “2.3 Statistical analysis” must be introduced in the revised version of the manuscript.   

6. No abbreviations should be used in the Legend of the Figures and Tables, as the reader has to understand the info without any reference to the main text. This comment applies to Legends of Figures 1, 4, 5, 6, 7, 8, 9 and Tables 3, 4 and 6.

7. What was the number of the Standard used by the authors to test the bioactivity of the resulting coatings (page 3, line 120)?

8. How did the authors succeed in pouring five droplets (6 μL each) on the same surface (i.e., 1 cm2) without influencing the measurements? Given the fact that the authors used small areas of 1 cm2, it is quite difficult, in this Reviewer’s opinion, to find five fresh zones in which to pour different drops. The authors have to explain this important detail.

9. The name of the first column in Tables 2, 3, 4, and 5 should read as “Samples code”. In addition, for Table 6, it should read as “Impedance spectroscopy/design parameters”.

10. “a, b, and c” should be denoted in the Legend of Figure 3 (page 7, lines 195 to 196).

11. In general, the used English is fine, but the authors should check again the whole manuscript for various typos. A few examples follow:

-        NH4F (page 3, line 105);

-        TiO2 (page 10, line 304);

-        TINT (page 10, line 306; page 12, line 355; page 14, line 417; page 15, line 459) ;

-        « profuced » (page 15, line 439).

In general, the used English is fine, but the authors should check again the whole manuscript for various typos. 

Reviewer 2 Report

The following points should be addressed:

1.    The authors could compare their results to the most famous kind of nanotubes, carbon nanotubes, which are also used in research and biocompatible films, like PEM thin films.1,2

2.    Introduction: The surface modification of titanium alloys with hydroxyapatite via MAO (PEO) is missing and should be added.3 Moreover, the surface modification with hydroxyapatite via Magnetron sputtering has been also reported.4

3.    Often space signs between a value and unit is missing and can be found throughout the whole manuscript. Please check it carefully.

4.    In chapter 2: Sometimes used devices are not correctly stated or information are missing. (type, company, city, country)

5.    Instead of the term Hanks’ solution, the term Hanks’ balanced salt solution (Hanks’ BSS) is familiar to me and it seems to me also more used by others. I suggest to change to this term.

6.    Figure 1: Caption – Do you mean TiNT? Please check it. Moreover, the unit for liters is here with lower case letter, but in the manuscript text correctly with upper case letter. Please correct this in Figure 1.

7.    Figure 2 and 6: The unit brackets are here [ ], but in the manuscript text and Table 1 is used ( ). Please change to the style in Figure 2 to unify it throughout the whole manuscript.

8.    Table 3 and 5: The unit style should be in the same way like in the rest of the manuscript. Please correct this. Moreover, it should be clarified what Sp. 1-8 means here. In table 5, space signs are missing. EDX mapping should be also provided to show the effective filling of the tubes.

9.    Figure 4: (a) and (c), the images are overexposed. Please provide better ones.

10. Figure 5: Scale bar style is different to the ones used in the other figures. Please unify it.

11. According to Figure 5, after the Hanks’ BSS has been applied. The morphology changed dramatically. The changes of the surface area should be calculated. Here also the question about the roughness is raising up. The Wenzel roughness should be considered or the roughness change in general. These is important, since the roughness and surface area are not looking suitable for cell settlement. Therefore, osseointegration could be a problem. However, the data are missing for a suitable discussion and evaluation.

12. Figure 6: It is common to use the lower case greek letter for an angle, like it is stated in line 129 or line 269. Please correct this. Moreover, the degree symbol should be used as well - „°“ (U+00B0).

13. For the XRD results. What about the Rutile phase? No amorphous titanium oxide is present in the samples? What is the amount of the different phases in the titanium oxide composition of the samples?

14. Wettability: It should be clarified that here are the water contact angles are presented. As the authors state that, the application range is wide. Nonpolar contact angles should be also provided, as well as the surface energy with all components. In this way, the authors will be able to discuss the settlement of bacteria/cells on the sample surface in a better way. Moreover, the osseointegration as stated by the authors can be better evaluated before real application.

References

(1)         Frueh, J.; Nakashima, N.; He, Q.; Moehwald, H. Effect of Linear Elongation on Carbon Nanotube and Polyelectrolyte Structures in PDMS-Supported Nanocomposite LbL Films. J Phys Chem B 2012, 116 (40), 12257–12262. https://doi.org/10.1021/jp3071458.

(2)         Voigt, J.; Christensen, J.; Shastri, V. P. Differential Uptake of Nanoparticles by Endothelial Cells through Polyelectrolytes with Affinity for Caveolae. Proc. Natl. Acad. Sci. U. S. A. 2014, 111 (8), 2942–2947. https://doi.org/10.1073/pnas.1322356111.

(3)         Kozelskaya AI, Rutkowski S, Frueh J, Gogolev AS, Chistyakov SG, Gnedenkov SV, Sinebryukhov SL, Frueh A, Egorkin VS, Choynzonov EL, et al. Surface Modification of Additively Fabricated Titanium-Based Implants by Means of Bioactive Micro-Arc Oxidation Coatings for Bone Replacement. Journal of Functional Biomaterials. 2022; 13(4):285. https://doi.org/10.3390/jfb13040285.

(4)         Kozelskaya AI, Fedotkin AY, Khlusov I, Litvinova L, Tverdokhlebov SI  Effect of working gas on physicochemical and biological properties of CaP coatings deposited by RFMS. Biomedical Materials. 2021, 16 (3), 035012. https://doi.org/10.1088/1748-605X/abcae3.

 Minor editing of English language required.

Round 2

Reviewer 1 Report

In my first letter, I have clearly presented important arguments against changing the title of the manuscript. Even though the authors added “biomedical applications” to the keywords, as suggested, they somehow “forgot” to amend the title correspondingly. I can understand the fact that deeper biological studies might be planned in the near future, but I can only discuss on the results presented in the current work. Therefore, the authors should understand this issue and must change the title to “Synthesis of titanium oxide nanotubes loaded with hydroxyapatite” in order for the manuscript to be accepted for publication.  

Regarding the statistical analysis, one can only assume that no statistical significance was observed between the investigated samples, since no “*” is present in the revised version of the manuscript. The authors must explain this and introduce a paragraph in the main text.

In Figure 7, in the case of S0 and S1 samples, there is no reason to put “0.0 ± 0.0”, since no result was obtained. The authors should correct this.  

The used English needs some minor corrections. The authors should therefore read carrefully the whole manuscript and perform all necessary corrections.

Reviewer 2 Report

The authors of the manuscript entitled “Synthesis of titanium oxide nanotubes loaded by hydroxyapatite for biomedical applications.” addressed and discussed the most of my comments in an appropriate way. After revision, the quality of the manuscript has increased. 

Minor editing of English language required

Author Response

Thank you for positive feedback. The necessary corrections to English grammar have been made and highlighted in yellow.